# Longitudinal electrophysiological changes after mesenchymal stem cell transplantation in a spinal cord injury rat model

Yuyo Maeda[1]*, Masaaki Takeda[1], Takafumi Mitsuhara[1], Takahito Okazaki[1], Kiyoharu Shimizu[1], Masashi Kuwabara[1], Masahiro Hosogai[1], Louis Yuge[2‡], Nobutaka Horie[1‡]

1 Department of Neurosurgery, Graduate School of Biomedical and Health Sciences, Hiroshima University, Hiroshima, Japan, 2 Division of Bio-Environmental Adaptation Sciences, Graduate School of Biomedical and Health Sciences, Hiroshima University, Hiroshima, Japan

☯ These authors contributed equally to this work.
‡ LY and NH also contributed equally to this work.
* maeday@hiroshima-u.ac.jp

**Data Availability Statement:** All relevant data are within the paper and its Supporting Information files.

## Abstract

Transcranial electrically stimulated motor-evoked potentials (tcMEPs) are widely used to evaluate motor function in humans and animals. However, the relationship between tcMEPs and the recovery of paralysis remains unclear. We previously reported that transplantation of mesenchymal stem cells to a spinal cord injury (SCI) rat model resulted in various degrees of recovery from paraplegia. As a continuation of this work, in the present study, we aimed to establish the longitudinal electrophysiological changes in this SCI rat model after mesenchymal stem cell transplantation. SCI rats were established using the weight-drop method. The model rats were transvenously transplanted with two types of mesenchymal stem cells (MSCs), one derived from rat cranial bones and the other from the bone marrow of the femur and tibia bone, 24 h after SCI. A phosphate-buffered saline (PBS) group that received only PBS was also created for comparison. The degree of paralysis was evaluated over 28 days using the Basso–Beattie–Bresnahan (BBB) scale and inclined plane task score. Extended tcMEPs were recorded using a previously reported bone-thinning technique, and the longitudinal electrophysiological changes in tcMEPs were investigated. In addition, the relationship between the time course of recovery from paralysis and reappearance of tcMEPs was revealed. The appearance of the tcMEP waveform was earlier in MSC-transplanted rats than in PBS-administered rats (earliest date was 7 days after SCI). The MEP waveforms also appeared at approximately the same level on the BBB scale (average score, 11 points). Ultimately, this study can help enhance our understanding of the relationship between neural regeneration and tcMEP recording. Further application of tcMEP in regenerative medicine research is expected.

**Funding:** This work was supported in part by a Grant-in-Aid for Scientific Research from the Japan Society for the Promotion of Science (JSPS KAKENHI grant number 18K16561 and 21K16609). The funders had no role in study design, data collection and analysis, decision to publish, or preparation of the manuscript.

**Competing interests:** The authors have declared that no competing interests exist.

## Introduction

Motor-evoked potentials (MEPs) are used to evaluate the function of the descending motor pathway by stimulating the motor cortex. Recent studies have indicated that the transcranial electrically stimulated MEP (tcMEP) is a useful tool for evaluating the function of the descending motor pathway and neurological dysfunction in an animal model of central nervous system (CNS) disorder [1–4]. The tcMEP has also been applied in clinical research in humans, particularly for intraoperative monitoring [5, 6] during neurosurgery [7, 8], spinal surgery [9, 10], and aortic surgery [11]. Moreover, the tcMEP has been reported as a predictor of postoperative neurological complications following aneurysm clipping surgery [7, 8] and neurologic outcomes in spine deformity surgery [10]. Reports on intraoperative monitoring revealed that a reduced MEP amplitude can be used to predict postoperative paralysis [12, 13]. Furthermore, some studies have provided mechanistic insights into changes in MEP expression after spinal cord injury (SCI), including studies in rats [14, 15]. However, it is not clear whether tcMEP recording can indicate recovery from paralysis in humans or other animals. Moreover, the correlation between the MEP waveform and degree of paralysis in rats has not been fully clarified.

We previously established a technique to record tcMEPs in rat models [16]. However, tcMEPs must be recorded in rats with varying levels of paralysis to investigate the above correlation over a relatively short period. We also reported that the transplantation of rat bone marrow-derived mesenchymal stem cells (rbMSCs) or rat cranial bone-derived mesenchymal stem cells (rcMSCs) to rat models of acute SCI enables the observation of the course of improvement in paralysis over time [17]. This previous study further showed that the waveform of tcMEPs disappears immediately after SCI and reappears with the recovery of motor function within a short period of 28 days [17]. Based on these findings, in the present study, we examined the relationship between the longitudinal electrophysiological changes and recovery from paralysis after the transplantation of MSCs in an SCI rat model.

## Materials and methods

### Ethics statement

All study protocols were approved by the Animal Experiment Committee of Hiroshima University, Japan, conducted under Project License A19-73. All methods have been reported in accordance with the ARRIVE guidelines (https://arriveguidelines.org), the ethical guidelines of the International Council for Laboratory Animal Science, and the Regulations for Animal Experiments, Hiroshima University.

### Animals

In the present study, we used female rats to account for the risk of bladder injury after SCI. Adult female Sprague–Dawley rats (10–11 weeks old, specific pathogen-free) with a mean body weight of 270 g (range, 250–300 g) were provided by Hiroshima University Animal Research Committee.

Breeding colonies were maintained in individually ventilated cages (Tecniplast, Buguggiate, Italy). The maximum caging density was three rats per cage. The rats were provided *ad libitum* access to food and water and were housed in a room with a 12-h/12-h light/dark cycle with controlled temperature (21˚C–25˚C) and humidity (40%–60%). Per the guidelines, animal husbandry and care were conducted in accordance with the current best practice, and all individuals involved in the care and use of the animals were trained and skilled to an acceptable level of competency. Euthanasia was performed according to the current best practice. Furthermore, appropriate anesthesia and analgesia were used to minimize pain and distress.

## Isolation and culture of rbMSCs and rcMSCs

rbMSCs and rcMSCs were isolated and cultured according to the methods described in our previous studies [18, 19]. Briefly, we collected the femur, tibia, and cranial bones of adult female Sprague–Dawley rats to isolate the rbMSCs and rcMSCs. Four rats were used for cell preparation. The cells were cultured in dishes (Sumitomo Bakelite Co., Tokyo, Japan) containing low-glucose Dulbecco's modified Eagle's medium (Sigma-Aldrich, St. Louis, MO, USA) supplemented with 10% fetal bovine serum (Thermo Fisher Scientific, Waltham, MA, USA) at 37°C under 5% $CO_2$. The culture medium was changed every 3 days.

## Characterization of rbMSCs and rcMSCs

To ensure the validity of the MSCs established in this study, their characteristics were analyzed using flow cytometry to detect the expression of cell surface markers and differentiation of MSCs as described previously [17, 19]. Cells were seeded at a plating density of 5000 cells per $cm^2$. After the rbMSCs and rcMSCs at passage 3 reached confluency, they were collected for flow cytometric analysis of MSC-specific markers and differentiation experiments. The expression of MSC markers, namely CD29, CD90, and CD44, and hematopoietic cell markers, namely CD34 and CD45, was investigated. IgG1 isotype control antibodies were used as negative controls. Data acquisition and analyses were conducted with biological replicates (different samples for each cell type) using a FACSVerse system (BD Biosciences, Franklin Lakes, NJ, USA). To test the cell differentiation ability of the mesodermal lineage, the rbMSCs and rcMSCs at passage 3 were allowed to differentiate into osteoblasts or adipocytes.

Osteogenic differentiation of MSCs was induced by culturing the cells in MSC osteogenic differentiation medium (PromoCell, Heidelberg, Germany) for 21 days according to the manufacturer's differentiation medium protocol, and the medium was changed every 3 or 4 days. Alizarin red S (Sigma-Aldrich) was used for final staining. For adipogenic differentiation, MSCs were cultured in MSC adipogenic differentiation medium (PromoCell) for 14 days according to the manufacturer's differentiation medium protocol, and the medium was changed every 3 or 4 days. Oil red O solution (Wako Pure Chemical Industries, Osaka, Japan) was used for final staining.

## Surgical procedure

Adult female Sprague–Dawley rats were used to construct an SCI model using the drop-weight method [20]. General anesthesia was induced in the rats using 3% isoflurane. A midline linear incision was made, and laminectomy of the Th10 vertebra was carried out. In this study, we used an originally created SCI machine with a modified rat stereotaxic apparatus and an original impactor. We have used the same machine in previously reported studies on rats with a SCI [17, 19, 21]. An impactor rod was set on the surface of the spinal cord at the Th10 vertebra, and a cylindrical brass weight (10 g) was dropped on the impactor. A spinal cord contusion was made with a calculated force of 100–150 kdyn. The rats that underwent the surgical procedure were administered antibiotics (cefazolin sodium: 20 mg/kg) for 5 days postoperatively to prevent infection. The bladders of SCI rats were compressed manually twice daily until sufficient recovery of autonomic bladder function. The SCI rats received rehabilitation, which also involved passive joint motion exercises daily for two weeks after SCI to prevent hindlimb joint contracture.

## Cell transplantation

Thirty-four female adult Sprague–Dawley rats were used in this study. The rats were randomly divided into four groups, and rats in all groups were subjected to skull thinning to record the

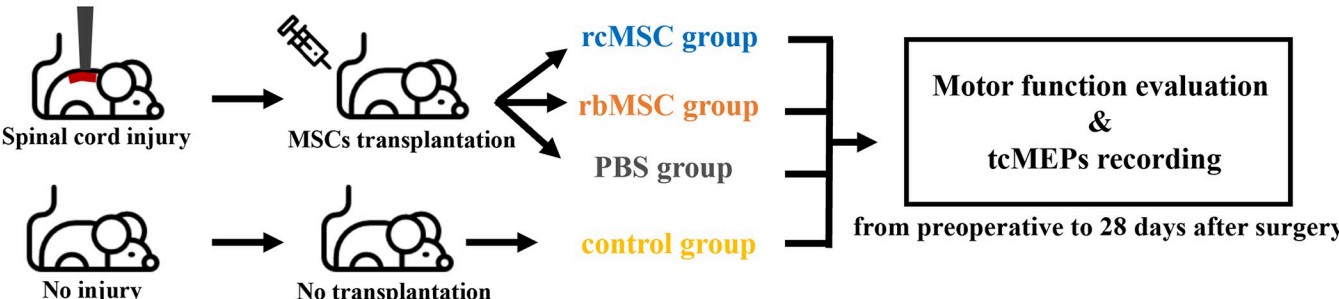

**Fig 1. Outline of the present experiment.** The rats were subjected to random grouping and a repeated-measures behavioral and electrophysiological study protocol from the preoperative stage to 28 days after surgery. We used the Basso–Beattie–Bresnahan scale and inclined plane task score as a behavioral endpoint and transcranial electrically stimulated motor-evoked potential (tcMEP) as an electrophysiological endpoint of neurological function recovery.

tcMEPs as described previously [16] (Fig 1). First, a control group of rats was established without SCI (n = 4), and only skull thinning was performed in this group. Further, rats with SCI were divided into the following three groups: PBS group (rats that received phosphate-buffered saline [PBS] only), rbMSC group (rats transplanted with rbMSCs), and rcMSC group (rats transplanted with rcMSCs; n = 10 each). At 24 h after SCI, $1.0 \times 10^6$ MSCs were transplanted into the rats of the rbMSC and rcMSC groups via the tail vein. Considering the similar number of animals in previous studies on MSCs [3, 18], our sample size was deemed to be sufficient to assess our model.

## Evaluation of paralysis after SCI

The Basso–Beattie–Bresnahan (BBB) scale [20, 22, 23] and inclined plane task score [2] were used to evaluate the degree of paralysis after SCI. The BBB scale is a 22-point scale that systematically follows recovery of hind limb function [20]. The inclined plane test is used to assess the maximum angle at which the rats can hold their position for 5 s on the inclined plane [2]. Paralysis was assessed on day 0 (immediately before surgery) and postoperative days 1, 7, 14, 21, and 28. All evaluations were performed by an observer blinded to the group identities.

## Recording of tcMEPs

The tcMEPs were recorded using a bone-thinning technique [16]. To prepare for tcMEP recording, the skulls of the rats were thinned using a diamond drill at the bregma and lambda positions with the rats under inhalation anesthesia with 1.5% isoflurane. Skull thinning was performed up to a diameter of 5 mm. The outer plate and diploe of the skull were removed, leaving only the inner plate. A train of four stimuli was administered for a 0.5 ms; tcMEP was recorded from the needle electrodes inserted in the quadriceps femoris of the rat hindlimb. Stimulation and recordings were performed using a Nicolet Endeavor CR intraoperative monitoring system (Nicolet Biomedical, Madison, WI, USA). The tcMEPs were recorded before surgery as the reference value and then on postoperative days 1, 7, 14, 21, and 28 in all groups. Skull thinning was performed only at the first recording, and no additional skull thinning was performed thereafter. Amplitude and latency were evaluated from the tcMEP waveforms at each time point and compared between the groups. All recordings were performed by an observer blinded to the group identities.

## Statistical analyses

Statistical analyses were performed using JMP software from SAS (version 15; SAS Institute, Cary, NC, USA). The period until the waveform appeared after SCI and the degree of paralysis

when the waveform appeared were compared using one-way analysis of variance (ANOVA) and post-hoc Tukey's honestly significant difference (HSD) test. Two-way analysis of variance and post-hoc Tukey's honestly significant difference test were used to analyze motor function and the amplitude of tcMEPs. Results with P < 0.05 were considered significant.

## Results

### Characterization of rat MSCs

Cell surface markers of isolated cells from the femur, tibia, and cranial bones of adult female Sprague–Dawley rats were analyzed using flow cytometry to confirm the cell identity as rbMSCs and rcMSCs. The results were positive for cell surface markers of MSCs, such as CD29, CD90, and CD44; in contrast, they were negative for cell surface markers of hematopoietic cells, such as CD34 and CD45 (S1 Table). To analyze mesodermal lineage cell differentiation, we investigated the osteogenic and adipocytic cells differentiated from isolated rbMSCs and rcMSCs. Cells positive for Alizarin red S and Oil red O staining were observed after differentiation (S1 Fig). The rbMSCs and rcMSCs showed characteristics similar to those reported previously [18, 19].

### Evaluation of paralysis after SCI

After establishing the SCI model using the drop-weight method [20, 21], paralysis was assessed using the BBB scale and inclined plane task scores. No significant decrease in these tests scores was observed in the control group (no SCI, only skull thinning). In contrast, the rats injected with PBS, rbMSCs, and rcMSCs showed severe paraplegia after SCI, with the scores improving over time. The rcMSC group showed the greatest recovery of motor function among the three groups after day 7 (P < 0.05; Fig 2A and 2B). The recovery of motor function in the rbMSC group was also higher than that in the PBS groups after day 7 (P < 0.05; Fig 2A and 2B).

### Recording of tcMEPs

Representative tcMEP waveforms are shown in Fig 3A and 3B. The control group showed no significant change in the MEP waveform over time (Fig 3A). In contrast, in the PBS, rbMSC, and rcMSC groups, the tcMEP waveform disappeared immediately after SCI. Reappearance of

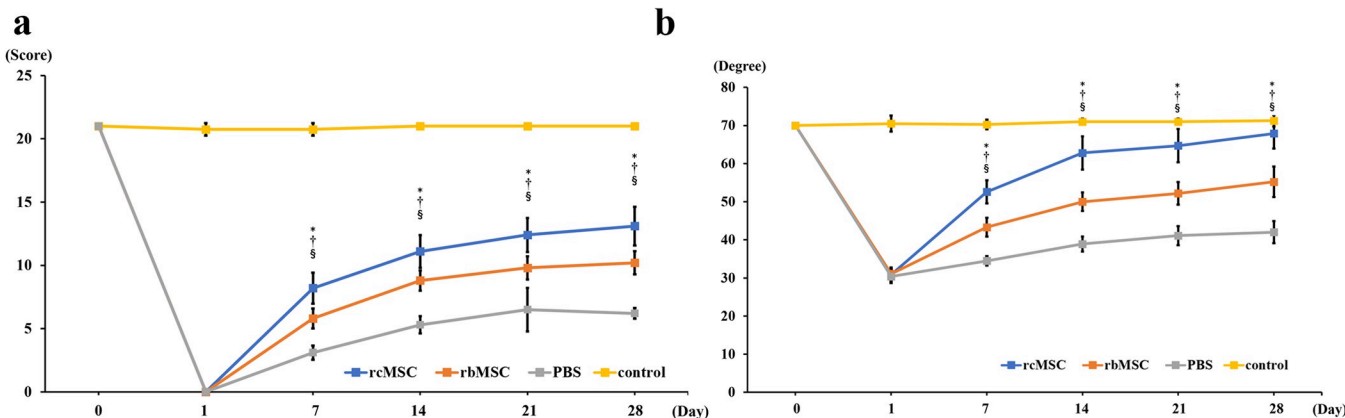

**Fig 2. Motor function recovery after spinal cord injury. a.** Results of the Basso–Beattie–Bresnahan (BBB) scale and **b.** inclined plane task score. Data are presented as mean ± standard deviation; n = 4, 10, 10, and 10 for the control, phosphate buffered saline (PBS), rat bone marrow-derived mesenchymal stem cell (rbMSC), and rat cranial bone-derived mesenchymal stem cell (rcMSC) groups, respectively. *P < 0.05, PBS group vs. rcMSC group; †P < 0.05, PBS group vs rbMSC group; §P < 0.05, rcMSC group vs. rbMSC group.

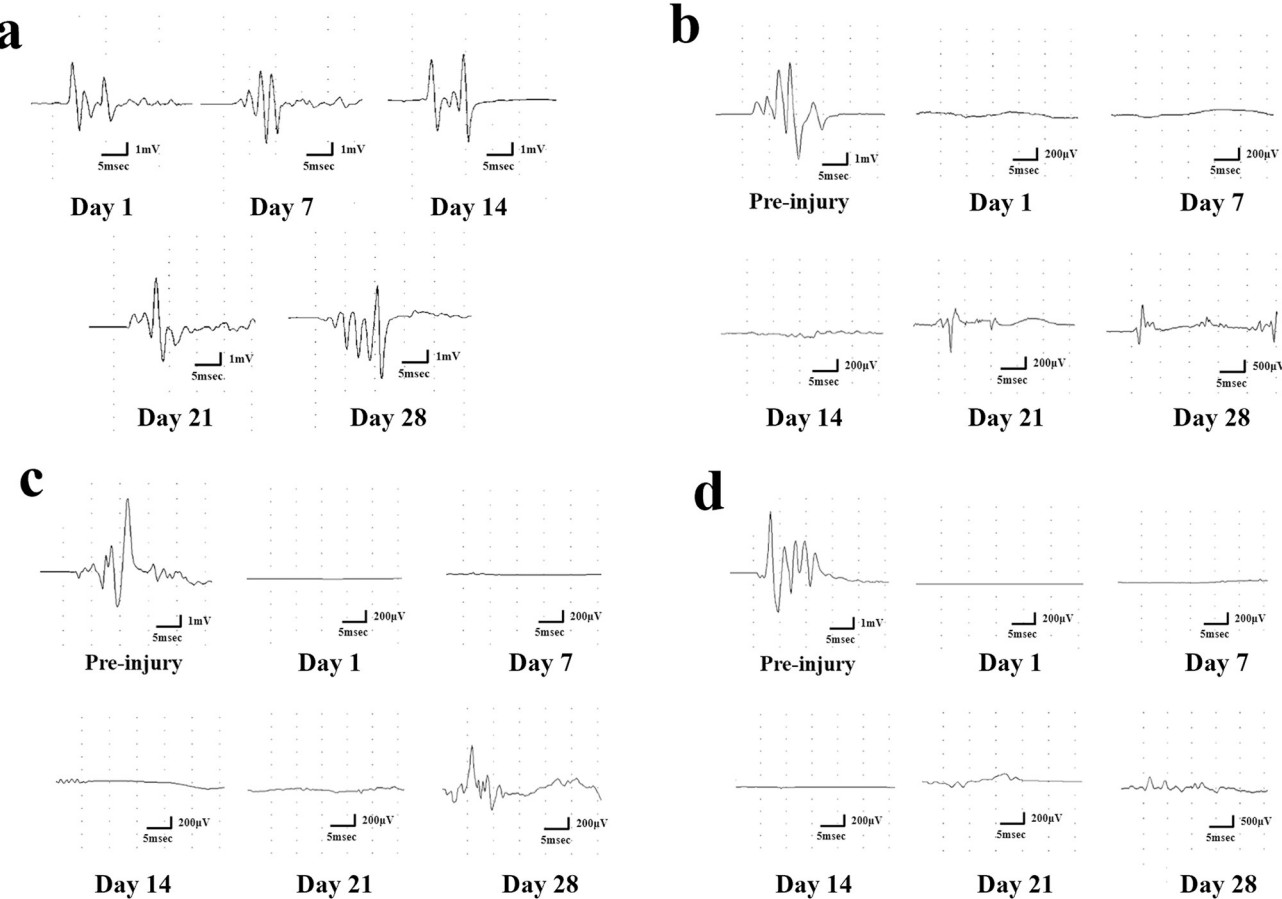

**Fig 3. Representative waveforms of transcranial electrically stimulated motor-evoked potentials (tcMEPs). a.** Representative waveforms of tcMEPs in control rats on days 1, 7, 14, 21, and 28. **b, c, d.** Representative reappearing waveform of tcMEPs in spinal cord injury (SCI) rats at preinjury, and on days 1, 7, 14, 21, and 28 in the phosphate buffered saline (PBS), rat bone marrow-derived mesenchymal stem cell (rbMSC), and rat cranial bone-derived mesenchymal stem cell (rcMSC) groups.

tcMEPs was confirmed in 17 of the 30 rats with SCI. Fig 3B–3D show the specific recovery process of tcMEP in the rcMSC, rbMSC, and PBS groups. Fig 4 shows the mean onset amplitude of each group. An increase in the amplitude over time was observed in all groups. The mean onset amplitudes in the rcMSC group were 102 ± 106.1 µV on day 7, 5651 ± 407.4 µV on day 14, 737.8 ± 957.2 µV on day 21, and 1599 ± 1173 µV on day 28; no waveform was recorded on day 1. The mean onset amplitudes in the rbMSC group were 356 ± 103.2 µV on day 21 and 496.7 ± 280.4 µV on day 28; no waveforms were recorded on days 1, 7, and 14. The onset amplitudes in the PBS group were 223 µV on day 21 and 461 µV on day 28; no waveforms were recorded on days 1, 7, and 14. In addition, the amplitudes in the rcMSC group were higher than that in other groups on days 14 and 28 (P < 0.05, Fig 4).

## Examination of the relationship between the time course of recovery from paralysis and reappearance of the tcMEP waveform

We investigated when the waveform first appeared, to examine the relationship between the time course of recovery from paralysis and reappearance of the tcMEP waveform. Fig 5A shows the number of rats evaluated at the first recording of tcMEP after SCI at each recording point. Based on the period from the induction of SCI to the reappearance of tcMEPs, two, six,

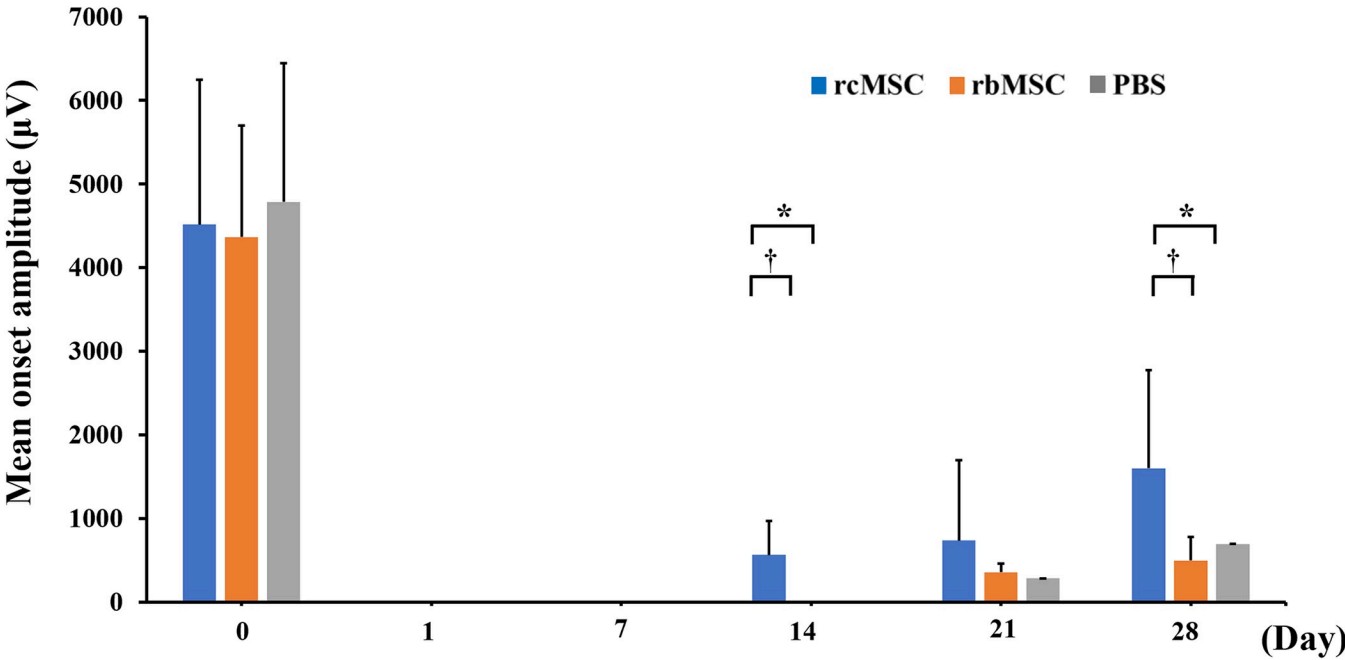

**Fig 4. Mean onset amplitude of each group.** The mean amplitude values showed an increase over time for each group. Data are presented as mean ± standard deviation; n = 10 for each group. *P < 0.05, phosphate buffered saline (PBS) group vs. rat cranial bone-derived mesenchymal stem cell (rcMSC) group; †P < 0.05, rcMSC group vs rat bone marrow-derived mesenchymal stem cell (rbMSC) group.

six, and three rats exhibited the phenomenon of the MEP waveform disappearing after SCI on days 7, 14, 21, and 28, respectively. In all groups, tcMEPs reappeared after an average of 15 days. Based on the presence or absence of cell transplantation among the 17 rats, MEP reappeared in 10 rats in the rcMSC group, 6 in the rbMSC group, and 1 in the PBS group. In addition, tcMEPs reappeared after an average of 14 and 25 days in the rcMSC and rbMSC groups,

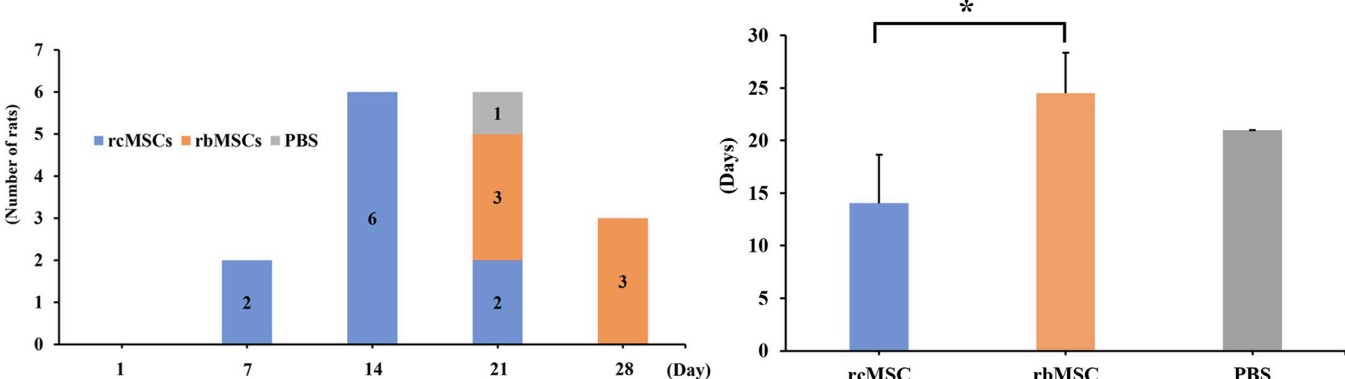

**Fig 5. Relationship between the time course of recovery from paralysis and reappearance of the transcranial electrically stimulated motor-evoked potential (tcMEP) waveform. a.** Number of rats at the first recording of tcMEP after spinal cord injury (SCI). Reappearance of tcMEPs was confirmed in 17 rats. In two, six, and two rats on days 7, 14, and 21, respectively, the first recorded tcMEPs after SCI were confirmed in the rat cranial bonel-derived mesenchymal stem cell (rcMSC) group. In three rats on days 21 and 28, the first recorded tcMEPs after SCI were confirmed in the rat bone marrow-derived mesenchymal stem cell (rbMSC) group. In the PBS group, the tcMEP reappeared in only one rat on day 21 after SCI. **b.** Number of days till the first recording of tcMEP after SCI. Data are presented as mean ± standard deviation; n = 10, 6, and 1 for rcMSC, rbMSC, and PBS groups, respectively. *P < 0.05, rcMSC group vs. rbMSC group.

respectively. In the PBS group, tcMEPs reappeared after 21 days, but only in one rat. The mean period until time at the first recording of tcMEPs after SCI was 14 ± 4.67 days in the rcMSC group, 24.5 ± 3.83 days in the rbMSC group, and 21 days in the PBS group. The period in the rcMSCs group was significant shorter than that in the rbMSCs group (Fig 5B).

### Examination of the relationship between the recovery of paralysis and reappearance of the tcMEP waveform

We investigated the degree of paralysis when the waveform appeared, to examine the relationship between the recovery of paralysis and reappearance of the tcMEP waveform. The BBB scale and inclined plane task score were used to assess paralysis.

The mean score for the BBB scale when the waveform appeared was 11.10 ± 0.938 (95% confidence interval 10.59–11.52, range: 10–13) (Fig 6A). However, there were no statistically significant differences in the BBB scale scores between the three groups. The mean threshold of the inclined plane task score when the waveform appeared was 58.06 ± 7.524 (95%

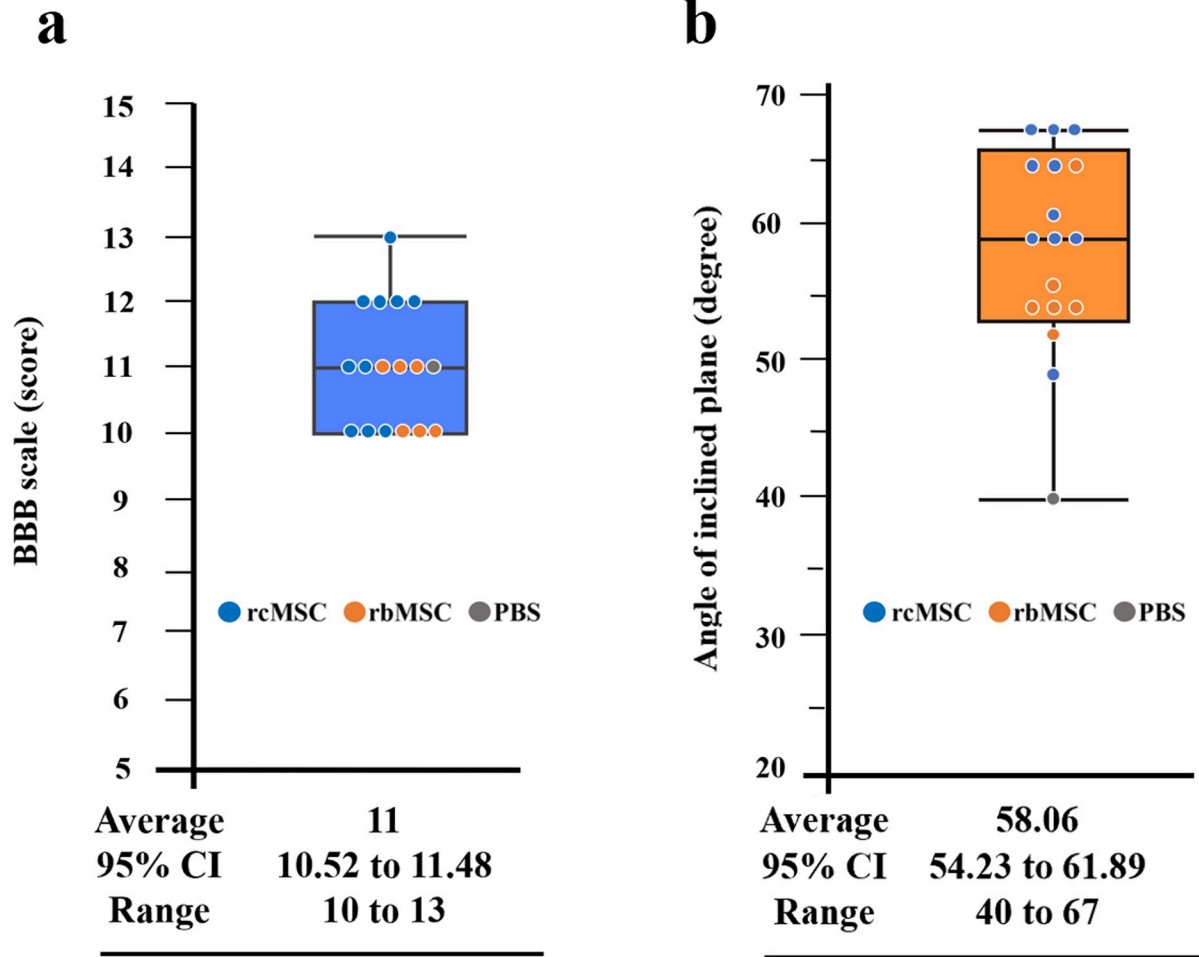

**Fig 6. Relationship between the recovery for paralysis and reappearance of the transcranial electrically stimulated motor-evoked potential (tcMEP) waveform. a.** Threshold Basso–Beattie–Bresnahan (BBB) scale and **b.** threshold inclined plane task score. Data are presented as mean ± standard deviation and 95% confidence interval (CI). Results for the rat cranial bone-derived mesenchymal stem cell (rcMSC), rat bone marrow-derived mesenchymal stem cell (rbMSC), and phosphate buffered saline (PBS) groups are indicated by blue, orange, and gray circles, respectively.

confidence interval 54.23–61.89, range: 40–67) (Fig 6B). However, there were no statistically significant differences in the inclined plane task scores between the three groups.

## Discussion

In the present study, the tcMEPs of the hindlimbs of rats were successfully recorded over a period of 1 month in the control group, and reappearance of tcMEPs was observed over time in some rats with SCI. tcMEP recordings are widely employed in both a clinical environment [5, 6] and in animal experiments [24–27]. For example, it has been used for neurophysiological evaluation in a CNS disorder model to analyze pyramidal tracts and the recovery of neural connections or regeneration of nerve fibers [3]. Previous studies involving intraoperative monitoring have demonstrated that reduced MEP amplitude is a predictor of postoperative paralysis [12, 13]. However, tcMEPs are difficult to record in humans with severe paralysis [12, 28]. Human studies in which tcMEPs were recorded excluded patients with severe paralysis with manual muscle test scores [29] of less than two [6]. However, in humans and animals, there is little evidence indicating that recovery from paralysis enables the recording of tcMEPs, and there are only a few reports of tcMEPs recorded over a period of several weeks in animal experiments [16, 22, 25]. In addition, the relationship between the degree of paralysis and tcMEP recording has not been determined yet. Even in rats, the threshold of paralysis for recording tcMEPs is still unknown. Here, we identified the correlation between the BBB scale and tcMEP recordings, and tcMEPs were found to reappear with a BBB scale of 10–11 points or higher after SCI. The results did not differ among the different origins of the transplanted MSCs. Moreover, we revealed the correlation between paralysis and tcMEPs. The tcMEPs reappeared with a BBB scale of 10–11 points or higher after SCI, indicating occasional to consistent supported plantar steps without forelimb–hindlimb coordination, with a score of 10 points or higher indicating that the rats could walk [20]. Our results suggest that the ability to walk serves as a cutoff for recording tcMEPs.

All 17 rats in which MEP reappeared showed mild paralysis after recovery. For the inclined plane task score, the 95% confidence interval of the threshold for recording tcMEPs was 54˚–64˚, although with variable results. The inclined plane task score results are affected by the function of both hindlimbs and forelimbs [30]. In addition to the degree of motor paralysis, the ability to prevent falls and stabilize the trunk affects these results [31]. These factors may explain the wide range of the results of the inclined plane task score in our study. Our results also suggest that the BBB scale is a more useful indicator of the availability of tcMEP recordings for rats after SCI than the inclined plane task score. These results are consistent with previous evidence showing that the MEP cannot be recorded in cases of severe paralysis in humans [6].

Most studies of tcMEPs in rat models of SCI focused on the responses to regenerative medicine, such as cell administration [3, 31] and new therapeutic techniques [1, 27, 31, 32], and tcMEPs were recorded at only one point, as a physiological endpoint of the recovery process. When planning a protocol for such experiments, there is no indicator of when tcMEPs should be recorded. Thus, our results can be used as an indicator to determine the optimal time point for recording tcMEPs. There have been some recent reports of extended recording tcMEPs in rats with SCI [17, 22, 25]. Even in these studies, overlooking tcMEP recordings in rats with severe paralysis that have not recovered to the recording threshold may have contributed to simplification of the experimental protocol and reduced the burden of invasive examinations on the rats. Thus, our findings can be applied to reduce not only unnecessary tcMEP recordings in rats with severe paralysis but also the appearance of false-positive results [33] and tissue damage caused by an excessive stimulation intensity [34].

Interestingly, focusing on the content of cell administration in the present study, the rcMSC group displayed a higher number of tcMEP reappearances (10/17 cases) and shorter period between SCI and the reappearance of tcMEPs (14 days) than the PBS and rbMSC groups. In addition, the rcMSC group showed significantly larger amplitudes of the tcMEP waveform during the course than the other two groups. The period until the time of reappearance of the tcMEP waveform in the rcMSC group was significantly shorter than that in the rbMSC group.

A previous study on electrophysiology in rats has reported the involvement of extrapyramidal tracts, in addition to pyramidal tracts, in motor function [35]. Both pathways are also involved in the formation of the MEP waveform [36]. Our data suggest a higher recovery effect of rcMSCs on the neural connections and regeneration of nerve fibers, playing a role in conduction [3] in pyramidal and extrapyramidal tracts than that of rbMSCs, and substantiates the findings of our previous study [17]. We have previously reported that the expression levels of the neurotrophic factors such as GDNF, BDNF, NGF, and VEGF were significantly higher in the rcMSCs than that in the rbMSCs [17, 19]. These factors suppress the apoptotic pathway and necroptosis pathway [37–39]. Previous studies have also reported that cranial bone-derived MSCs showed a more effective nursing effect on the abovementioned two cell death pathways in neurons than bone marrow-derived MSCs [17, 18]. In the present study, these nursing effects of neurotrophic factors may have reduced secondary damage by suppressing cell death pathways in cases of acute SCI and consequently contributed to electrophysiological recovery. Furthermore, rcMSCs expressing higher levels of neurotrophic factors may have a stronger electrophysiologic improvement effect in pyramidal and extrapyramidal tracts. Previous studies have already reported that rcMSCs transplantation reduces cavity formation more effectively and is more involved in the regeneration of connections between neurons in the spinal cord after SCI than rbMSCs [17, 19]. These differences in the recovery effects on the spinal cord between rcMSCs and rbMSCs may have resulted in the differences in electrophysiological functional recovery in the present study. Further studies are warranted to elucidate the relationship between these effects of MSC transplantation and nerve regeneration in the spinal cord.

The present study had a limitation. Most of the rats that showed reappearance of tcMEP waveform in the present study were MSC-transplanted rats. Based on the relationship between paralysis and tcMEP recordings obtained in the present study, it is difficult to generalize this result as threshold paralysis for recording tcMEPs after SCI. In the future, the accumulation of tcMEP recordings in more mild SCI rats without MSC transplantation will contribute to the clarification of threshold paralysis for recording tcMEPs that are yet to be clarified. As a future application of tcMEPs, we will attempt to record tcMEPs in more models of CNS injury, investigate the relationship between tcMEP recording and long-term neurological prognosis, and determine its usefulness as a predictor of long-term prognosis for CNS disorder models.

## Conclusion

We determined the longitudinal electrophysiological changes in an SCI rat model transplanted with MSCs. Revealing the time course of recovery from paralysis and paralysis at the time of tcMEP reappearance can help enhance understanding of the relationship between neural regeneration and tcMEP recording. Furthermore, this study demonstrated the value of tcMEPs in evaluating nerve function recovery in regenerative medicine research.

## Supporting information

**S1 Table. Surface marker expression in mesenchymal stem cells (MSCs) from various tissues.**
(DOCX)

**S1 Fig. Lineage-specific assessment following the differentiation of rat bone marrow-derived mesenchymal stem cells (rbMSC) and rat cranial bone-derived mesenchymal stem cells (rcMSC).** (A) Alizarin red S staining following the osteogenic differentiation of rbMSCs and rcMSCs. (B) Oil red O staining following the adipogenic differentiation of rbMSCs and rcMSCs. Scale bars, 50 μm.
(DOCX)

## Acknowledgments

The authors wish to thank Reo Kawano for helpful discussions and comments on the manuscript. This work is supported by TWO CELLS Co., Ltd., Hiroshima, Japan.

## Author Contributions

**Conceptualization:** Yuyo Maeda, Takafumi Mitsuhara, Takahito Okazaki, Kiyoharu Shimizu.

**Data curation:** Yuyo Maeda, Masaaki Takeda.

**Formal analysis:** Yuyo Maeda.

**Funding acquisition:** Masaaki Takeda, Takafumi Mitsuhara.

**Investigation:** Yuyo Maeda.

**Methodology:** Yuyo Maeda, Masaaki Takeda, Kiyoharu Shimizu, Masashi Kuwabara, Masahiro Hosogai, Louis Yuge.

**Project administration:** Yuyo Maeda.

**Supervision:** Nobutaka Horie.

**Validation:** Masaaki Takeda, Louis Yuge, Nobutaka Horie.

**Visualization:** Yuyo Maeda, Nobutaka Horie.

**Writing – original draft:** Yuyo Maeda.

**Writing – review & editing:** Louis Yuge, Nobutaka Horie.

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
