## [Decision Letter · Decision Letter 0]

16 Jun 2022

PONE-D-22-10569Longitudinal electrophysiological changes after mesenchymal stem cell transplantation in a spinal cord injury rat modelPLOS ONE

Dear Dr. Maeda,

Thank you for submitting your manuscript to PLOS ONE. After careful consideration, we feel that it has merit but does not fully meet PLOS ONE’s publication criteria as it currently stands. Therefore, we invite you to submit a revised version of the manuscript that addresses the points raised during the review process.

We look forward to receiving your revised manuscript.

Kind regards,

Sujeong Jang

Academic Editor

PLOS ONE

Journal Requirements:

Reviewers' comments:

Reviewer's Responses to Questions

**Comments to the Author**

1. Is the manuscript technically sound, and do the data support the conclusions?

Reviewer #1: Partly

Reviewer #2: Yes

2. Has the statistical analysis been performed appropriately and rigorously? 

Reviewer #1: No

Reviewer #2: Yes

3. Have the authors made all data underlying the findings in their manuscript fully available?

Reviewer #1: Yes

Reviewer #2: Yes

4. Is the manuscript presented in an intelligible fashion and written in standard English?

Reviewer #1: No

Reviewer #2: Yes

5. Review Comments to the Author

Reviewer #1: PONE-D-22-10569

Longitudinal electrophysiological changes after mesenchymal stem cell transplantation in a spinal cord injury rat model

Maeda and colleagues reported in this manuscript that transcranial electrically stimulated motor evoked potentials (tcMEPs) evaluated the functional improvement after intravenous infusion of rat bone marrow-derived mesenchymal stem cells (rbMSCs) or rat skull-derived mesenchymal stem cells (rcMSCs) in a rat model of contused spinal cord injury (SCI). This study is potentially interesting, however, there are several issues to be clarified.

Methods

1. Animals

Something wrong with the body weight. Female SD rats (4-5 weeks) were used in this study and the authors stated a mean body weight is 270g (250-300g). According to the Charles River website, body weight is less than 100g until 5 weeks. Please clarify.

2. Isolation and culture of rbMSCs and exMSCs

How many animals did the authors use for cell preparation?

3. Characterization of……………

For the differentiation experiments, detail should be presented. How many cells? How was the plating density? How long did their culture? Etc….

4. Surgical procedure

Detail should be presented in terms of SCI machine. Did the author use the NYU impactor? Detailed information on the antibiotics should be provided. The SCI rats received rehabilitation. The standard protocol should be provided.

Results

1. Recording of tcMEPs

Line 212. Fig2a and b should be Fig3a and b.

2. Fig 4, 5, 6. These data do not provide anything scientifically meaningful results. The authors also did not perform a statistically analyzed.

The authors also did not perform a statistically analyzed.

Reviewer #2: This is an interesting and novel study focusing on the effect of two mesenchymal stem cell lines (rbMSCs and rcMSCs) on functional recovery after spinal cord contusion injury. While the study is relatively well designed and performed there are few points to be answered and expanded in detail.

Minor points:

line 123: Th10 should preferably read Th10 vertebra

line 125: the force applied by the weight drop instument should be expressed in cN or kdyn, g/cm is not relevant here.

line 138: rbMSCs and rcMSCs are bit confusing at first glance. What does "rc" mean? One would expect the abbreviation "rs" (from rat skull), which help the reader to easily distinguish between the treatment groups.

Major points:

There is considerable difference in the regeneration-promoting capacity of the two cell lines. What is the scenario? Please provide the reader with more details which could explain the different functional improvement achieved by the two cell lines and discuss these in detail. As there are no morphological investigations supporting the functional findings, the explanations should be very realistic. Please note that the corticospinal tract does not play as important role in the functional improvement of rodents, esp. that of rats as in the case of humans. Please discuss other alternatives.

6. PLOS authors have the option to publish the peer review history of their article (what does this mean?). If published, this will include your full peer review and any attached files.

Reviewer #1: No

Reviewer #2: No

---

## [Author Response · Author response to Decision Letter 0]

25 Jun 2022

manuscript ID: PONE-D-22-10569

Longitudinal electrophysiological changes after mesenchymal stem cell transplantation in a spinal cord injury rat model

Point-by-point response to the reviewers’ comments

Reviewer #1: 

Maeda and colleagues reported in this manuscript that transcranial electrically stimulated motor evoked potentials (tcMEPs) evaluated the functional improvement after intravenous infusion of rat bone marrow-derived mesenchymal stem cells (rbMSCs) or rat skull-derived mesenchymal stem cells (rcMSCs) in a rat model of contused spinal cord injury (SCI). This study is potentially interesting, however, there are several issues to be clarified.

Our Response: 

Thank you for reviewing our manuscript and for your positive feedback and comments.

Methods

1. Animals

Something wrong with the body weight. Female SD rats (4-5 weeks) were used in this study and the authors stated a mean body weight is 270g (250-300g). According to the Charles River website, body weight is less than 100g until 5 weeks. Please clarify.

Our Response: 

Thank you for your important comment. We have checked with Charles River to confirm the correct age of the rats. The weight is unchanged, but the age is 10–11 weeks. The text has also been corrected. The corrections are highlighted in yellow in the revised manuscript (Page 4, lines 85–88).

2. Isolation and culture of rbMSCs and exMSCs

How many animals did the authors use for cell preparation?

Our Response: 

In this study, four rats were used for cell preparation. This information was added to the Methods sections. The corrections are highlighted in yellow in the revised manuscript (Page 5, lines 103–104).

3. Characterization of……………

For the differentiation experiments, detail should be presented. How many cells? How was the plating density? How long did their culture? Etc…. 3. 3. 

Our Response: 

Thank you very much for your comment. We have added the details of the experimental procedure regarding the differentiation experiment. We seeded cells at a plating density of 5000 cells/cm2 and the number of cells to be seeded in culture. As for the culture period, we started the differentiation experiment when the cells reached confluency. The culture period for the differentiation experiments was performed according to the differentiation medium protocol created by the manufacturer. This information has been added to the Methods sections. The corrections are highlighted in yellow in the revised manuscript (Pages 4–5, lines 112–130).

4. Surgical procedure

Detail should be presented in terms of SCI machine. Did the author use the NYU impactor? 

Our Response: 

In our laboratory, we used a self-made SCI machine and impactor, which is a modified rat stereotaxic instrument. We have used a similar machine and have reported studies using rats with spinal cord injury (1,2,3,4). This information has been added to the Methods sections. The corrections are highlighted in yellow in the revised manuscript (Page 6, lines 135–138).

(1) Mitsuhara T. et al. Simulated microgravity facilitates cell migration and neuroprotection after bone marrow stromal cell transplantation in spinal cord injury. Stem Cell Res Ther. 4,　(2013). 

(2) Imura, T. et al. Hypoxic Preconditioning Increases the Neuroprotective Effects of Mesenchymal Stem Cells in a Rat Model of Spinal Cord Injury. Stem Cell Res Ther. 7, (2017).

(3) Otsuka T. et al. Comparisons of neurotrophic effects of mesenchymal stem cells derived from different tissues on chronic spinal cord injury rats. Stem Cells Dev. 30: 865-875, (2021).

(4) Maeda Y. et al. Transplantation of rat cranial bone-derived mesenchymal stem cells promotes functional recovery in rats with spinal cord injury. Sci Rep. 11, (2021).

Detailed information on the antibiotics should be provided. 

Our Response: 

In the present study, rats were treated with a dose of cefazolin sodium (20 mg/kg). This information has been added to the Methods sections. The corrections are highlighted in yellow in the revised manuscript (Page 6, line 141).

The SCI rats received rehabilitation. The standard protocol should be provided.

Our Response: 

Rehabilitation was performed in the present study. The details and duration of this program have been added to the Methods section. The corrections are highlighted in yellow in the revised manuscript (Pages 6–7, lines 144–145).

Results

1. Recording of tcMEPs

Line 212. Fig2a and b should be Fig3a and b.

Our Response:

Thank you for pointing this out. The error has been revised per your comment.

2. Fig 4, 5, 6. These data do not provide anything scientifically meaningful results. The authors also did not perform a statistically analyzed. 

Our Response:

Thank you for your valuable suggestion. We have added the statistical analysis for Figures 4, 5, and 6. One additional figure has been added as Figure 5 due to the additional analysis. The Methods, Result, and Discussion sections were revised accordingly with the additional analysis. The corrections are highlighted in yellow in the revised manuscript.

Reviewer #2: 

This is an interesting and novel study focusing on the effect of two mesenchymal stem cell lines (rbMSCs and rcMSCs) on functional recovery after spinal cord contusion injury. While the study is relatively well designed and performed there are few points to be answered and expanded in detail.

Our Response: 

Thank you for reviewing our manuscript and for your positive feedback and comments.

Minor points:

line 123: Th10 should preferably read Th10 vertebra

Our Response:

Thank you for pointing this out. The text has been corrected per your comment.

line 125: the force applied by the weight drop instument should be expressed in cN or kdyn, g/cm is not relevant here.

Our Response:

The force applied by the weight-drop instrument has been calculated. The units have been changed to kdyn and have been added to the Methods section.

line 138: rbMSCs and rcMSCs are bit confusing at first glance. What does "rc" mean? One would expect the abbreviation "rs" (from rat skull), which help the reader to easily distinguish between the treatment groups.

Our Response:

We apologize for the confusion. We made a mistake in the Introduction section. The abbreviation “rcMSCs” stands for “rat cranial bone-derived MSCs.” In all previous reports from our laboratory (1，2，3，4), we have used the term "cranial bone-derived" to refer to MSCs derived from the skull. We use the same terminology in the present study. We have revised the text accordingly. The corrections are highlighted in yellow in the revised manuscript (Page 3, line 68).

(1) Abiko M, Mitsuhara. et al. Rat cranial bone-derived mesenchymal stem cell transplantation promotes functional recovery in ischemic stroke model rats. Stem Cells Dev. 27: 1053-1061, (2018). 

(2) Shinagawa K. et al. The characteristics of human cranial bone marrow mesenchymal stem cells. Neurosci Lett. 606: 161-166, (2015).

(3) Otsuka T. et al. Comparisons of neurotrophic effects of mesenchymal stem cells derived from different tissues on chronic spinal cord injury rats. Stem Cells Dev. 30: 865-875, (2021).

(4) Maeda Y. et al. Transplantation of rat cranial bone-derived mesenchymal stem cells promotes functional recovery in rats with spinal cord injury. Sci Rep. 11, (2021).

Major points:

There is considerable difference in the regeneration-promoting capacity of the two cell lines. What is the scenario? Please provide the reader with more details which could explain the different functional improvement achieved by the two cell lines and discuss these in detail. As there are no morphological investigations supporting the functional findings, the explanations should be very realistic. Please note that the corticospinal tract does not play as important role in the functional improvement of rodents, esp. that of rats as in the case of humans. Please discuss other alternatives.

Our Response:

Thank you for your important comment. We have previously reported that cranial bone-derived MSCs express higher levels of neurotrophic factors than bone marrow-derived MSCs (1). We considered that the anti-cell death effect mediated by this high expression of neurotrophic factor contributed to the restoration of electrophysiological nerve function (2). In addition, cranial bone-derived MSCs have been shown to be more effective than bone marrow-derived MSCs in suppressing cavity formation and improving nerve connections in spinal cord injury (1, 2). We hypothesized that these differences in transplantation efficacy between the two types of MSCs may have resulted from this perspective. Furthermore, we hypothesized that the difference in these transplantation effects between the two types of MSCs may have resulted in the difference in the electrophysiological recovery process as shown in the present study. These details have been added to the Discussion section. The corrections are highlighted in yellow in the revised manuscript.

In addition, it has been reported that not only the extrapyramidal tracts but also the pyramidal tracts are involved in the motor function of rats (3). Furthermore, both pathways are involved in the formation of MEP waveforms (4). This information has been added to the Discussion section with appropriate citations. The corrections are highlighted in yellow in the revised manuscript.

(1) Otsuka T. et al. Comparisons of neurotrophic effects of mesenchymal stem cells derived from different tissues on chronic spinal cord injury rats. Stem Cells Dev. 30: 865-875, (2021).

(2) Maeda Y. et al. Transplantation of rat cranial bone-derived mesenchymal stem cells promotes functional recovery in rats with spinal cord injury. Sci Rep. 11, (2021).

(3) Shiau JS, Zappulla RA, Nieves. The effect of graded spinal cord injury on the extrapyramidal and pyramidal motor evoked potentials of the rat. Neurosurgery. 1992 Jan;30(1):76-84.

(4) Konrad PE, Tacker WA. Pyramidal versus extrapyramidal origins of the motor evoked potential. Neurosurgery. 1991 Nov;29(5):795-6.

---

## [Decision Letter · Decision Letter 1]

21 Jul 2022

Longitudinal electrophysiological changes after mesenchymal stem cell transplantation in a spinal cord injury rat model

PONE-D-22-10569R1

Dear Dr. Maeda,

We’re pleased to inform you that your manuscript has been judged scientifically suitable for publication and will be formally accepted for publication once it meets all outstanding technical requirements.

Kind regards,

Sujeong Jang

Academic Editor

PLOS ONE

Additional Editor Comments (optional):

Reviewers' comments:

Reviewer's Responses to Questions

**Comments to the Author**

1. If the authors have adequately addressed your comments raised in a previous round of review and you feel that this manuscript is now acceptable for publication, you may indicate that here to bypass the “Comments to the Author” section, enter your conflict of interest statement in the “Confidential to Editor” section, and submit your "Accept" recommendation.

Reviewer #1: All comments have been addressed

Reviewer #2: All comments have been addressed

2. Is the manuscript technically sound, and do the data support the conclusions?

Reviewer #1: Yes

Reviewer #2: Yes

3. Has the statistical analysis been performed appropriately and rigorously? 

Reviewer #1: Yes

Reviewer #2: Yes

4. Have the authors made all data underlying the findings in their manuscript fully available?

Reviewer #1: Yes

Reviewer #2: Yes

5. Is the manuscript presented in an intelligible fashion and written in standard English?

Reviewer #1: Yes

Reviewer #2: Yes

6. Review Comments to the Author

Reviewer #1: The authors have adequately addressed my comments raised in a previous round of review and I feel that this manuscript is now acceptable for publication.

Reviewer #2: (No Response)

7. PLOS authors have the option to publish the peer review history of their article (what does this mean?). If published, this will include your full peer review and any attached files.

Reviewer #1: No

Reviewer #2: No

---

## [Editor Report · Acceptance letter]

28 Jul 2022

PONE-D-22-10569R1 

Longitudinal electrophysiological changes after mesenchymal stem cell transplantation in a spinal cord injury rat model 

Dear Dr. Maeda:

I'm pleased to inform you that your manuscript has been deemed suitable for publication in PLOS ONE. Congratulations! Your manuscript is now with our production department. 

Kind regards, 

on behalf of

Dr. Sujeong Jang 

Academic Editor

PLOS ONE